# GENERATIVE PDE CONTROL

**Long Wei**[1][*], **Peiyan Hu**[2][*][§], **Ruiqi Feng**[1][*], **Yixuan Du**[3][§], **Tao Zhang**[1], **Rui Wang**[4][§],
**Yue Wang**[5], **Zhi-Ming Ma**[2], **Tailin Wu**[1][†]
[1]School of Engineering, Westlake University,
[2]Academy of Mathematics and Systems Science, Chinese Academy of Sciences,
[3]Jilin University,  [4]Fudan University,  [5]Microsoft Research Asia
 weilong@westlake.edu.cn, hupeiyan18@mails.ucas.ac.cn
 fengruiqi@westlake.edu.cn, wutailin@westlake.edu.cn

## ABSTRACT

Controlling PDE is a fundamental task across science and engineering. Classical techniques for PDE control tend to be computationally demanding and recent deep learning-based approaches often struggle to optimize long-term control sequences. In this work, we introduce Diffusion generative PDE Control (DiffConPDE), a new class of method to address the PDE control problem. DiffConPDE excels by simultaneously minimizing both the learned generative energy function and the predefined control objectives across the entire trajectory and control sequence. Moreover, we enhance DiffConPDE with prior reweighting, enabling the discovery of control sequences that significantly deviate from the training distribution. We test our method in 2D jellyfish movement in a fluid environment and 1D Burgers' equation control. Our method consistently outperforms baselines. Notably, DiffConPDE unveils an intriguing fast-close-slow-open pattern observed in the jellyfish, aligning with established findings in the field of fluid dynamics.

## 1 INTRODUCTION

The PDE control problem injects time-variant signals to steer evolution and optimize specific objectives for a PDE. It is a fundamental task with applications, such as including controlled nuclear fusion Degrave et al. (2022), fluid control Holl et al. (2020), and chemical engineering Christofides & Chow (2002). Controlling high-dimensional, complex PDE systems in an efficient way presents three significant challenges. **(1) Simulation challenge.** The physical system is typically high-dimensional and highly nonlinear. **(2) Optimization challenge.** The PDE control task requires to optimize over a highly non-convex, potentially high-dimensional control sequence on top of the physical simulation. **(3) Partial observation and control.** In practical applications, our ability to observe or exert control over the physical system is often constrained to a limited subset of its components.

To tackle PDE control problems, various techniques have been proposed. Classical methods typically adopt numerical simulation, among which the adjoint method Lions (1971) is widely applied but computationally expensive. Regarding traditional control methods, Model Predictive Control (MPC) Schwenzer et al. (2021) is limited by high computational costs and challenges in optimizing for a globally optimal solution. Recent advances in deep learning have impressive performance. But both supervised learning (SL) Holl et al. (2020); Hwang et al. (2022) and reinforcement learning (RL) Farahmand et al. (2017); Pan et al. (2018) may fall into adversarial modes Zhao et al. (2022b) and struggle to optimize long-term control sequence. See Appendix A for more related work.

In this work, we introduce Diffusion generative PDE Control (DiffConPDE), a *new class* of method to address the PDE control problem. We take an energy optimization perspective to implicitly capture the constraints inherent in system dynamics through the diffusion model trained using system trajectory data and control sequences. This prevents the generated system dynamics from falling out of distribution, and offers an enhanced perspective of optimization overlong-term dynamics.

An essential aspect of PDE control lies in its capacity to generate near-optimal controls. We address this challenge with the key insight that the learned generative energy landscape can be *decomposed*

---

[*]Equal contribution. [§]Work done as an intern at Westlake University. [†]Corresponding author.

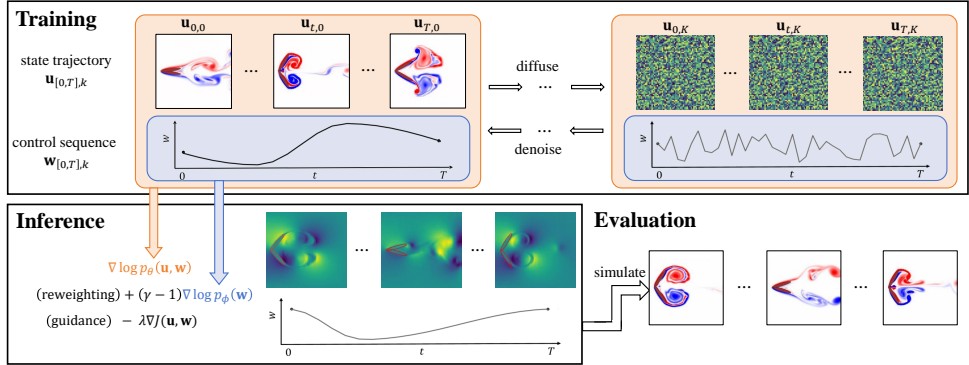

Figure 1: **Overview of DiffConPDE**. The figure depicts the training (top), inference (bottom left), and evaluation (bottom right) of DiffConPDE. Orange and blue colors respectively represent models learning the joint distribution $p_\theta(\mathbf{u}, \mathbf{w})$ and the prior distribution $p_\phi(\mathbf{w})$.

into two components: a prior distribution representing the control sequence and a conditional distribution characterizing the system trajectories given the control sequence. We then introduce a *prior reweighting* technique to DiffConPDE, which constitutes the our second key contribution.

We demonstrate the effectiveness of DiffConPDE via extensive experiments on 2D Jellyfish flapping and 1D Burgers' equation control problems. Our method consistently outperforms widely-applied classical control methods and state-of-the-art deep learning-based methods.

## 2 METHOD

### 2.1 PROBLEM SETUP

We consider the following widely applicable PDE systems with external control signals:

$$\frac{\partial \mathbf{u}}{\partial t} + \mathcal{F}(\mathbf{u}, \nabla \mathbf{u}, \nabla^2 \mathbf{u}, \mathbf{w}) = 0 \tag{1}$$

$$\mathcal{B}(\mathbf{u}, \nabla \mathbf{u})|_{\mathbf{x} \in \partial\Omega} = 0 \tag{2}$$

$$\mathbf{u}|_{t=0} = \mathbf{u}_0. \tag{3}$$

Here, $\mathbf{u}(t, \mathbf{x}) : [0, \mathcal{T}] \times \Omega \mapsto \mathbb{R}^{d_\mathbf{u}}$ is the trajectory of the system defined on time range $[0, \mathcal{T}] \subset \mathbb{R}$ and spatial domain $\Omega \subset \mathbb{R}^D$. Similarly, $\mathbf{w}(t, \mathbf{x}) : [0, \mathcal{T}] \times \Omega \mapsto \mathbb{R}^{d_\mathbf{w}}$ is the external control signal. $\mathcal{F}$ is an operator that characterizes the dynamics of the PDE system. Eq. (2) is the boundary condition where $\mathcal{B}$ is a linear operator operating on the boundary $\partial\Omega$. The initial condition is specified by $\mathbf{u}_0(\mathbf{x})$. The control objective is $\mathcal{J}(\mathbf{u}, \mathbf{w})$, whose minimization defines the PDE control task as

$$\mathbf{w}^* = \arg\min_{\mathbf{w}} \mathcal{J}(\mathbf{u}, \mathbf{w}) \quad \text{s.t.} \quad \mathcal{C}(\mathbf{u}, \mathbf{w}) = 0, \tag{4}$$

where $\mathcal{C}(\mathbf{u}, \mathbf{w}) = 0$ includes the PDE dynamics Eq. (1), the boundary condition Eq. (2) and the initial condition Eq. (3). It is crucial to emphasize that in numerous scenarios, explicit PDE expressions are unattainable, and we can only have access to observed control sequences and trajectory data, by which the PDE constraints $\mathcal{C}(\mathbf{u}, \mathbf{w})$ are implicitly characterized.

### 2.2 GENERATIVE CONTROL BY DIFFUSION MODELS

We model the PDE constraints as a parameterized energy-based model (EBM) $E_\theta(\mathbf{u}, \mathbf{w}, \mathbf{c})$ which characterizes the distribution $p(\mathbf{u}, \mathbf{w}|\mathbf{c})$ conditioned on conditions $\mathbf{c}$ of the PDE by the correspondence $p(\mathbf{u}, \mathbf{w}|\mathbf{c}) \propto \exp(-E_\theta(\mathbf{u}, \mathbf{w}, \mathbf{c}))$. Then the problem Eq. (4) can be converted to:

$$\mathbf{u}^*, \mathbf{w}^* = \arg\min_{\mathbf{u},\mathbf{w}} [E_\theta(\mathbf{u}, \mathbf{w}, \mathbf{c}) + \lambda \cdot \mathcal{J}(\mathbf{u}, \mathbf{w})], \tag{5}$$

where $\lambda$ is a hyperparameter. This formulation optimizes $\mathbf{u}$ and $\mathbf{w}$ of all time steps simultaneously.

**Training.** We train a diffusion model to estimate $E_\theta$. The denoising network (Ho et al., 2020) in the diffusion model is trained using $\mathcal{L} = \mathbb{E}_{k \sim U(1,K),(\mathbf{z},\mathbf{c}) \sim p(\mathbf{z},\mathbf{c}),\boldsymbol{\epsilon} \sim \mathcal{N}(\mathbf{0},\mathbf{I})}[\|\boldsymbol{\epsilon} - \boldsymbol{\epsilon}_\theta(\tilde{\mathbf{z}}, \mathbf{c}, k)\|_2^2]$, where $\mathbf{z}$ denotes the concatenation of $\mathbf{u}$ and $\mathbf{w}$ and $\tilde{\mathbf{z}} := \sqrt{\bar{\alpha}_k}\mathbf{z} + \sqrt{1 - \bar{\alpha}_k}\boldsymbol{\epsilon}$. The training dataset $p(\mathbf{z}, \mathbf{c})$ is either simulated using a numerical solver or collected from observation date in realistic systems.

---

**Algorithm 1** Inference for DiffConPDE

---

1: **Require** Diffusion models $\epsilon_\theta(\mathbf{z}_k, \mathbf{c}, k)$ and $\epsilon_\phi(\mathbf{w}_k, \mathbf{c}, k)$, control objective $\mathcal{J}(\cdot)$, covariance matrix $\sigma_k^2 I$, control conditions $c$, schedule $\bar{\alpha}_k$, hyperparameters $\lambda, \gamma, K$
2: Initialize optimization variables $\mathbf{z}_K \sim \mathcal{N}(\mathbf{0}, \mathbf{I})$
3: **for** $k = K, \ldots, 1$ **do**
4:    $\xi_1, \xi_2 \sim \mathcal{N}\left(0, \sigma_k^2 \mathbf{I}\right)$
5:    $\hat{\mathbf{z}}_k = (\mathbf{z}_k - \sqrt{1 - \bar{\alpha}_k}\epsilon_\theta(\mathbf{z}_k, \mathbf{c}, k))/\sqrt{\bar{\alpha}_k}$
6:    $\mathbf{z}_{k-1} = \mathbf{z}_k - \eta(\epsilon_\theta(\mathbf{z}_k, \mathbf{c}, k) + \lambda\nabla_{\mathbf{z}}\mathcal{J}(\hat{\mathbf{z}}_k)) + \xi_1$
7:    $\mathbf{w}_{k-1} = \mathbf{w}_{k-1} - \eta(\gamma - 1)\epsilon_\phi(\mathbf{w}_k, \mathbf{c}, k) + \xi_2$
8: **end for**
9: $\mathbf{u}^*, \mathbf{w}^* = \mathbf{z}_0$
10: **return** $\mathbf{u}^*, \mathbf{w}^*$

---

**Control optimization.** After $\epsilon_\theta$ is trained, Eq. (5)) can be optimized by the Langevin sampling procedure as follows. We start from an initial sample $\mathbf{z}_K \sim \mathcal{N}(\mathbf{0}, \mathbf{I})$, and run the process $\mathbf{z}_{k-1} = \mathbf{z}_k - \eta\left(\nabla_{\mathbf{z}}(E_\theta(\mathbf{z}_k, \mathbf{c}) + \lambda\mathcal{J}(\hat{\mathbf{z}}_k)\right) + \xi$ iteratively, where $\xi \sim \mathcal{N}\left(\mathbf{0}, \sigma_k^2 \mathbf{I}\right)$. Here $\hat{\mathbf{z}}_k$ is the approximate noise-free $\mathbf{z}_0$ estimated from $\mathbf{z}_k$ by $\hat{\mathbf{z}}_k = (\mathbf{z}_k - \sqrt{1 - \bar{\alpha}_k}\epsilon_\theta(\mathbf{z}_k, \mathbf{c}, k))/\sqrt{\bar{\alpha}_k}$, where $\bar{\alpha}_k := \prod_{i=1}^k \alpha_i$. The gradient of $E_\theta$ can then be replaced by our trained denoising network $\epsilon_\theta$ as follows:

$$\mathbf{z}_{k-1} = \mathbf{z}_k - \eta\left(\epsilon_\theta(\mathbf{z}_k, \mathbf{c}, k) + \lambda\nabla_{\mathbf{z}}\mathcal{J}(\hat{\mathbf{z}}_k)\right) + \xi, \quad \xi \sim \mathcal{N}\left(\mathbf{0}, \sigma_k^2 \mathbf{I}\right) \tag{6}$$

where $\sigma_k^2$ and $\eta$ correspond to noise schedules and scaling factors used in the diffusion process, respectively. Iteration of this denoising process for $k = K, K - 1, ..., 1$ yields a final solution $\mathbf{z}_0 = \{\mathbf{u}_{[1,T],0}, \mathbf{w}_{[0,T-1],0}\}$ for the optimization problem Eq. (5).

## 2.3 Prior Reweighting

In PDE control, a critical challenge lies in obtaining control sequences superior to those in training datasets. To mitigate impact of the prior distribution, we propose a *prior reweighting* technique, which introduces an adjustable hyperparameter $\gamma > 0$, allowing for tuning the influence of this prior distribution. We denote the reweighted version of $p(\mathbf{u}, \mathbf{w}|\mathbf{c})$ as $p_\gamma(\mathbf{u}, \mathbf{w}|\mathbf{c}) := p(\mathbf{w}|\mathbf{c})^\gamma p(\mathbf{u}|\mathbf{w}, \mathbf{c})/Z = p(\mathbf{w}|\mathbf{c})^{\gamma-1}p(\mathbf{u}, \mathbf{w}|\mathbf{c})/Z$, where $Z$ is a normalization constant and $0 < \gamma < 1$. Then the energy model $E^{(\gamma)}(\mathbf{u}, \mathbf{w}, \mathbf{c})$ that learns $-\log(p_\gamma(\mathbf{u}, \mathbf{w}|\mathbf{c}))$ can be decomposed as

$$E^{(\gamma)}(\mathbf{u}, \mathbf{w}, \mathbf{c}) = (\gamma - 1)E_\phi(\mathbf{w}, \mathbf{c}) + E_\theta(\mathbf{u}, \mathbf{w}, \mathbf{c}) - \log Z, \tag{7}$$

where $E_\theta(\mathbf{u}, \mathbf{w}, \mathbf{c})$ follows Eq. (5), and $E_\phi(\mathbf{w}, \mathbf{c})$ is another diffusion model that learns $-\log p(\mathbf{u} \mid \mathbf{c})$ which can be trained similarly to $E_\theta$. Then the optimization problem Eq. (5) can be transformed into

$$\mathbf{u}^*, \mathbf{w}^* = \underset{\mathbf{u}, \mathbf{w}}{\arg\min}\left[E^{(\gamma)}(\mathbf{u}, \mathbf{w}, \mathbf{c}) + \lambda \cdot \mathcal{J}(\mathbf{u}, \mathbf{w})\right]. \tag{8}$$

The optimization of the above problem leads to the following iteration in the diffusion model:

$$\mathbf{z}_{k-1} = \mathbf{z}_k - \eta(\epsilon_\theta(\mathbf{z}_k, \mathbf{c}, k) + \lambda\nabla_{\mathbf{z}}\mathcal{J}(\hat{\mathbf{z}}_k)) + \xi_1, \tag{9}$$

$$\mathbf{w}_{k-1} = \mathbf{w}_{k-1} - \eta(\gamma - 1)\epsilon_\phi(\mathbf{w}_k, \mathbf{c}, k) + \xi_2, \tag{10}$$

where $\xi_1, \xi_2 \sim \mathcal{N}\left(0, \sigma_k^2 \mathbf{I}\right)$, and $\mathbf{z}_k = [\mathbf{u}_k, \mathbf{w}_k]$. The overview of DiffConPDE is illustrated in Figure 1. When $\gamma = 1$, the model $\epsilon_\phi$ is not needed, and we denote this simplified version as DiffConPDE-lite.

## 3 Experiments

We conduct experiments on the vital and challenging 1D Burger's Equation control and 2D jellyfish movement control problems. For 1D Burgers' equation, we use baselines: (1) the adjoint method Lions (1971) with 10 and 100 time steps; (2) PID Li et al. (2006) interacting with the ground-truth solver and surrogate model; (3) an RL method named SAC Haarnoja et al. (2018) with online, offline and pseudo-online versions; and (4) the Supervised Learning method (SL) Hwang et al. (2022). Specifically, the online and pseudo-online version of SAC interacts with the solver and the surrogate model of the solver respectively, while the offline version only uses given data. For 2D jellyfish movement control, baselines include SAC (offline), SL, and MPC Schwenzer et al. (2021). The adjoint method and PID are inapplicable to this data-driven task. Since interaction with the solver is time-consuming, SAC (online) is not applied. Baselines are detailed in Appendix D and E.

Table 1: **Best $J_{\text{actual}}$ achieved in 1D Burgers's equation control.** Bold font denotes the best model, and underline denotes the second best model. Baselines trained with the ground-truth solver are segregated and placed at the top, as drawing a comparison with them would be inequitable.

|  | PO-FC | FO-PC | PO-PC |
|---|---|---|---|
| PID (solver) | - | 0.02305 | **0.00090** |
| SAC (online) | 0.01567 | 0.04334 | 0.02768 |
| Adjoint (10 steps) | - | 0.03251 | - |
| Adjoint (100 steps) | - | 0.02944 | - |
| PID (surrogate model) | - | 0.09115 | 0.09631 |
| SAC (pseudo-online) | 0.01577 | 0.03426 | 0.02149 |
| SAC (offline) | 0.03201 | 0.04333 | 0.03328 |
| SL | 0.09752 | 0.00078 | 0.02328 |
| **DiffConPDE-lite (ours)** | 0.01139 | **0.00037** | 0.00494 |
| **DiffConPDE (ours)** | **0.01103** | **0.00037** | 0.00494 |

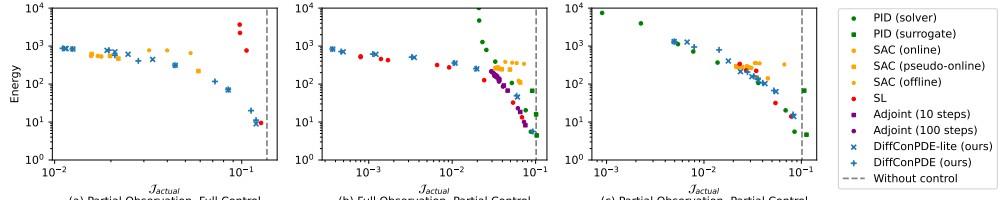

Figure 2: **Pareto frontier of energy vs. $J_{\text{actual}}$ of different methods for 1D Burgers' equation.**

## 3.1 1D BURGERS' EQUATION CONTROL

**Experiment settings.** The Burgers' equation is a fundamental PDE and we consider 1D Burgers' equation with the Dirichlet boundary condition and external force $w(t, x)$ with the following form:

$$\begin{cases} \frac{\partial u}{\partial t} = -u \cdot \frac{\partial u}{\partial x} + \nu \frac{\partial^2 u}{\partial x^2} + w(t, x) & \text{in } \Omega \times [0, T] \\ u(t, x) = 0 & \text{on } \partial\Omega \times [0, T] \\ u(0, x) = u_0(x) & \text{in } \Omega \times \{t = 0\}. \end{cases} \tag{11}$$

Here $\nu$ is viscosity, and $u_0(\mathbf{x})$ is the initial condition. The objective of control is to minimize

$$J_{\text{actual}} := \int_\Omega |u(T, x) - u_d(x)|^2 \mathrm{d}x \tag{12}$$

while constraining the energy cost of the control sequence

$$\int_{[0,T] \times \Omega} |w(t, x)|^2 \mathrm{d}t \mathrm{d}x \tag{13}$$

subject to Eq. (11), where $u_d(x)$ is the given target state.

We select three experiment settings that correspond to different real-life scenarios: partial observation, full control (PO-FC), full observation, partial control (FO-PC), and partial observation, partial control (PO-PC), which are detailed in Appendix B.2. It should be noted that online methods (PID and online SAC) have unfair advantages. Nevertheless, we included them here for the sake of clarity.

**Results.** In Table 1, we report results of the optimal control error $J_{\text{actual}}$ of different methods. It can be observed that our DiffConPDE delivers the best results when compared to all baselines except in PO-PC where PID (solver) has an unfair edge. DiffConPDE and DiffConPDE-lite show little performance gap since the prior distribution $p(w|u_0, u_T)$ is conditioned on both $u_0$ and $u_T$, which fully determines the optimal $w$. Thus, $p(w|u_0, u_T)$ is intrinsically the optimal distribution and there is no need to suppress it, allowing DiffConPDE-lite to deliver satisfactory results.

To compare the ability of different methods to optimize $J_{\text{actual}}$ with constrained energy cost, we compare the Pareto frontiers of different methods. As can be observed in Figure 2, the Pareto frontiers of DiffConPDE is consistently among the best, achieving the *lowest* $J_{\text{actual}}$ for most settings of the energy budget. Interestingly, although PID (solver) has an unfair advantage due to its interaction with the solver, it only slightly outperforms our method in the PO-PC setting (c), and requires orders of magnitude larger energy to achieve a better $J_{\text{actual}}$. Although SL performs well in full observation setting (b), it encounters difficulty in partial observation scenarios (a)(c). The results demonstrate DiffConPDE's advantage to generate near-optimal control sequences.

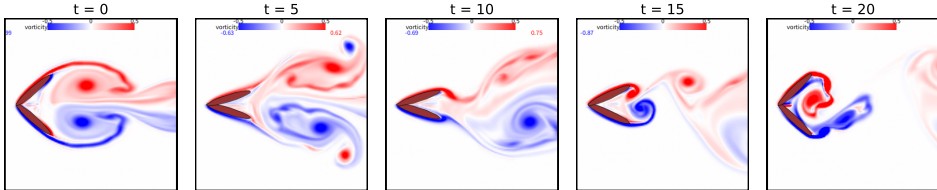

Figure 3: **Comparison of generated control curves of three test jellyfish.** The resulting control objective $\mathcal{J}$ for each curve is presented.

Figure 4: **Movement and fluid field visualization on the jellyfish controlled by DiffConPDE as in the middle subfigure of Figure 3.**

Table 2: **Results of 2D jellyfish movement control experiments.** Bold font denotes the best model.

|  | Full observation | | | Partial observation | | |
|---|---|---|---|---|---|---|
|  | $\bar{v} \uparrow$ | $R(\mathbf{w}) \downarrow$ | $\mathcal{J} \downarrow$ | $\bar{v} \uparrow$ | $R(\mathbf{w}) \downarrow$ | $\mathcal{J} \downarrow$ |
| MPC | 25.72 | 0.0112 | 109.17 | -150.51 | 0.1791 | 329.59 |
| SAC (pseudo-online) | -166.96 | 0.0069 | 18.14 | -153.09 | 0.0057 | 158.82 |
| SAC (offline) | -158.66 | 0.0069 | 165.58 | -206.21 | 0.0058 | 211.96 |
| SL | -76.94 | 0.1286 | 205.57 | -102.98 | 0.1188 | 221.79 |
| **DiffConPDE (ours)** | **279.87** | 0.2058 | **-74.11** | **150.21** | 0.1269 | **-23.32** |

## 3.2 2D Jellyfish Movement Control.

**Experiment settings.** The task is to control the movement of a flapping jellyfish with two wings in a 2D fluid field. The dynamic of fluid follows the 2D incompressible Navier-Stokes Equation:

$$\begin{cases} \frac{\partial \mathbf{v}}{\partial t} + \mathbf{v} \cdot \nabla \mathbf{v} - \nu \nabla^2 \mathbf{v} + \nabla p = 0 \\ \nabla \cdot \mathbf{v} = 0 \\ \mathbf{v}(0, \mathbf{x}) = \mathbf{v}_0(\mathbf{x}), \end{cases} \tag{14}$$

where $\mathbf{v}$ is the fluid's 2D velocity, and $p$ is the pressure, constituting the PDE state $\mathbf{u} = (\mathbf{v}, p)$. The initial velocity condition is $\mathbf{v}_0(\mathbf{x})$ and the kinematic viscosity is $\nu$. The jellyfish's boundary can be parameterized by the opening angle $\mathbf{w}_t$ of wings due to rigidity. Consequently, the control objective is to maximize its average moving speed $\bar{v}$, under the energy cost constraint $R(\mathbf{w})$ and the periodic constraint $d(\mathbf{w}_T, \mathbf{w}_0)$ of the movement:

$$\mathcal{J} = -\bar{v} + \zeta \cdot R(\mathbf{w}) + d(\mathbf{w}_T, \mathbf{w}_0), \tag{15}$$

subject to Eq. (14) and the boundary condition that velocity vanishes near the boundary. The hyperparameter $\zeta$ is set to be 1000. We evaluate in two settings: full observation, where the full state $\mathbf{u} = (\mathbf{v}, p)$ is observed; and partial observation, where only $p$ is observed. Details are in Appendix C.

**Results.** Evaluation results are presented in Table 2. Our method outperforms the baselines by a large margin in optimizing the control objective $\mathcal{J}$. Configuration of the hyperparameter $\gamma$ in DiffConPDE and performance with respect to varying $\gamma$ is presented in Appendix G. Even in the more challenging partial observation setting, DiffConPDE still exhibits substantial advantages over existing methods. This reflects our method has a strong capability to PDE control under inadequate information.

Figure 3 visualizes generated opening angle curves of different methods on three test jellyfish. Opening angle curves of DiffConPDE show an obvious fast-close-slow-open shape, which is proven to produce high speed in jellyfish movement Kang et al. (2023). While this mode of movement appears rarely in the training dataset, DiffConPDE could generate such control sequences for most test samples. This reflects that diffusion models under guidance and prior reweighting are effective in optimizing the control objective. Conversely, opening angles obtained by baselines are inferior. The movement and resulting fluid field of the jellyfish corresponding to the middle subfigure of Figure 3 controlled by DiffConPDE is illustrated in Figure 4.

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

## A  Additional Related Work

### A.1  PDE Simulation

PDE simulation forms the foundation of PDE control. While classical numerical techniques for simulating PDEs are renowned for their accuracy, they are often associated with significant computational expenses Morton & Mayers (2005); Lapidus & Pinder (1999). Recently, neural network-based PDE solvers show a significant advantage in accelerating PDE simulations. They could be roughly divided into three primary classes: data-driven methods Li et al. (2021); Sanchez-Gonzalez et al. (2020); Pfaff et al. (2020); Brandstetter et al. (2022); Wu et al. (2022); Brandstetter et al. (2023); Lam et al. (2023), Physics-Informed Neural Networks (PINNs) Raissi et al. (2019); Cai et al. (2021); Wang et al. (2021), and solver-in-the-loop methods Um et al. (2020); Vlachas et al. (2022). Most of them use an iterative horizontal prediction framework. Instead, we treat PDE trajectory as a whole variable and use diffusion models to learn an explicit PDE simulator conditioned on control sequences. A notable work is by Cachay et al. (2023), which introduces diffusion model for temporal forecasting. While both our work and Cachay et al. (2023)'s employ diffusion models, we tackle a different task of PDE control. Furthermore, we incorporate the control objective into the inference, and introduce prior reweighting to tune the influence of the prior.

### A.2  PDE Control

For decades, the adjoint method has been the most widely used approach for solving PDE control problems Lions (1971); McNamara et al. (2004); Protas (2008). It is accurate but computationally expensive. Deep learning-based methods have emerged as a powerful tool for modeling physical systems' dynamics. Holl et al. (2020) propose a hierarchical predictor-corrector scheme to control complex nonlinear physical systems over long time frames. A more recent work proposed by Hwang et al. (2022) designs two stages which respectively learn the solution operator and search for optimal control. Different from these methods, we do not use the surrogate model, and learn both state trajectories and control sequences in an integrated way. Reinforcement learning is also applied to control PDEs Pan et al. (2018) or fluid systems Larcher & Hachem (2022). Particularly in the field of fluid dynamics, reinforcement learning has been applied to a multitude of specific problems including drag reduction Rabault et al. (2019); Elhawary (2020), conjugate heat transfer Beintema et al. (2020); Hachem et al. (2021) and swimming Novati et al. (2017); Verma et al. (2018). But they implicitly consider physics information and sequentially make decisions. In contrast, we generalize the entire trajectories, which results in a global optimization with consideration of physical information learned by models.

### A.3  Diffusion Models

Diffusion models Ho et al. (2020) have significantly advanced in applications such as image and text generation Dhariwal & Nichol (2021); Nichol et al. (2022), inverse design Wu et al. (2024); Vlastelica et al. (2023), inverse problem Holzschuh et al. (2023), physical simulation Cachay et al. (2023); Price et al. (2023), and decision-making Janner et al. (2022); Ajay et al. (2022). Generating diverse yet consistent samples poses a challenge. For diversity, methods Liu et al. (2022); Bao et al. (2023); Zhao et al. (2022a); Du et al. (2023) that integrate score estimates from various models have been effective. For consistency, guidance diffusion techniques Dhariwal & Nichol (2021); Ho & Salimans (2021) have been utilized to generate condition-specific samples. Our approach differs by flattening the joint distribution to achieve better control by slightly expanding beyond the prior distribution range.

## B  Additional Details for 1D Burgers' Equation Control

### B.1  Data Generation

We use the finite difference method (called solver or ground-truth solver in the following) to generate the training data for the 1D Burgers' equation. Specifically, the initial value $\mathbf{u}_0(x)$ and the control sequence $\mathbf{w}(t, x)$ are both randomly generated, and then the states $\mathbf{u}(t, x)$ are numerically computed using the solver.

Figure 5: **Visualization of different types of observation in 1D Burgers' equation and performance of different methods on a testing sample.** In each of the three settings, $u$ at the final time step controlled by each method is shown for one sample in the testing set. Target denotes the control target state $u_d$.

In the numerical simulation (using the ground-truth solver), a domain of $x = [0, 1]$, $t = [0, 1]$ is simulated. The space is discretized into 128 grids and time into 10000 steps. However, in the dataset, only 10 time stamps are stored. For the control sequence $\mathbf{w}$, its refreshing rate is $0.1^{-1}$, i.e., $\mathbf{w}(t, x)$, $t \in [0.1k, 0.1(k + 1)]$, $k \in \{0, .., 9\}$ does not change with $t$. Therefore, the data size of each trajectory is $[10, 128]$ for the state $\mathbf{u}$ and $[11, 128]$ for the force $f$.

In all settings, the initial value $\mathbf{u}(0, x)$ is a superposition of two Gaussian functions $\mathbf{u}(0, x) = \sum_{i=1}^{2} a_i e^{-\frac{(x-b_i)^2}{2\sigma_i^2}}$, where $a_i, b_i, \sigma_i$ are all randomly sampled from uniform distributions: $a_1 \sim U(0, 2)$, $a_2 \sim U(-2, 0)$, $b_1 \sim (0.2, 0.4)$, $b_2 \sim (0.6, 0.8)$, $\sigma_1 \sim U(0.05, 0.15)$, $\sigma_2 \sim U(0.05, 0.15)$. Similarly, the control sequence $\mathbf{w}(x, t)$ is also a superposition of 8 Gaussian functions

$$\mathbf{w}(t, x) = \sum_{i=1}^{8} a_i e^{-\frac{(\mathbf{x}-b_{1,i})^2}{2\sigma_{1,i}^2}} e^{-\frac{(t-b_{2,i})^2}{2\sigma_{2,i}^2}}, \tag{16}$$

where each parameter is independently generated as follows: $b_{1,i} \sim U(0, 1)$, $b_{2,i} \sim U(0, 1)$, $\sigma_{1,i} \sim U(0.05, 0.2)$, $\sigma_{2,i} \sim U(0.05, 0.2)$, while $a_1 \sim U(-1.5, 1.5)$ and for $i \geq 2$, $a_i \sim U(-1.5, 1.5)$ or 0 with equal probabilities. $\mathbf{u}(t, x)$, $(t \neq 0)$ is then numerically simulated (using the ground-truth solver) given $\mathbf{u}(0, x)$ and $\mathbf{w}(t, x)$ based on Eq. (11). The setting of the dataset generation is based on a previous work (Hwang et al., 2022).

We generated 90000 trajectories for the training set and 50 for the testing set. Each trajectory takes up 32KB space and the size of the dataset sums up to 2GB.

## B.2 EXPERIMENTAL SETTING

We select three different experiment settings that correspond to different real-life scenarios: partial observation, full control (PO-FC), full observation, partial control (FO-PC), and partial observation, partial control (PO-PC), which are illustrated in Figure 5. These settings are challenging for classical control methods such as PID since they require capturing the long-range dependencies in the system dynamics. Note that the reported metrics $\mathcal{J}$ in different settings are not directly comparable. Following are three different settings of our experiments.

### B.2.1 PARTIAL OBSERVATION, FULL CONTROL

In realistic scenarios, the system is often unable to be observed completely. Generally speaking, it is impractical to place sensors *everywhere* in a system, so the ability of the model to learn from incomplete data is imperative. To evaluate this, we hide some parts of $\mathbf{u}$ in this setting and measure the $J_{\text{actual}}$ of model control.

Specifically, $\mathbf{u}(t, x)$, $x \in [\frac{1}{4}, \frac{3}{4}]$ is set to zero in the dataset during training and $\mathbf{u}_0(x)$, $x \in [\frac{1}{4}, \frac{3}{4}]$ is also set to zero during testing. In this partial observation setting $\Omega = [1, \frac{1}{4}] \cup [\frac{3}{4}, 1]$. Since no information in the central $\frac{1}{2}$ space is ever known, the model does not know what will influence the control outcome of the unobserved states. Therefore, controlling the unobserved states is not a reasonable task and they are excluded from the evaluation metric.

This setting is particularly challenging not only because of the uncertainty introduced by the unobserved states but also the generation of the control in the central locations that implicitly affect the controlled $\mathbf{u}$ at $x \in \Omega$.

### B.2.2  Full Observation, Partial Control

This is another setting of practical relevance, where only a fraction of the system can be controlled. The control sequence is enforced to be zero in the central locations of $x \in [\frac{1}{4}, \frac{3}{4}]$. $\Omega$ is still $[0, 1]$, and $\mathcal{J}$ is evaluated on all of the observed states, though.

Some modifications to the dataset should be mentioned. The generation of the data involves first generating $\mathbf{w}$ as before, followed by setting the central $\frac{1}{2}$ of $\mathbf{w}$ to zero. To compensate for the decreased control intensity so that the magnitude of $\mathbf{u}$ can be roughly comparable to the full control setting, we double the magnitude of $\mathbf{w}$. During the evaluation, the output control sequence is also post-processed to be zero in $x \in [\frac{1}{4}, \frac{3}{4}]$.

It is worth noting that in this setting, even when the control energy is not limited at all, it is still challenging to find a perfect control since the model has to learn how to indirectly impose control on the central locations.

### B.2.3  Partial Observation, Partial Control

The final setting is the combination of the previous two settings. Only $\Omega = [0, \frac{1}{4}] \cup [\frac{3}{4}, 1]$ is observed, controlled and evaluated.

It is worth noting that some models require accessing the current state to produce output. If the model interacts with the ground truth solver instead of a surrogate model, then the result would be unfairly good since the information of the unobserved states is leaked through the interaction.

### B.3  Guidance conditioning.

In addition to the introduced explicit guidance in the main text, conditioning is also widely used to guide sampling in diffusion models (Ho & Salimans, 2021; Shu et al., 2023). When the control objective can be naturally expressed in a conditioning form, e.g., the generated trajectory $\mathbf{u}$ is required to coincide with a desired target $\mathbf{u}^*$, we can include $\mathbf{u} = \mathbf{u}^*$ as a condition in $\mathbf{c}$ such that the sampled trajectory $\mathbf{u}$ from diffusion models automatically satisfying $\mathbf{u} = \mathbf{u}^*$. Overall, in our proposed DiffConPDE framework, the control objective $\mathcal{J}$ can be optimized either using the explicit guidance $\nabla \mathcal{J}$ or guidance conditioning depending on the specific control objectives.

### B.4  Model

Since the training of models $\boldsymbol{\epsilon}_\phi \approx \nabla_{\mathbf{w}} \log p(\mathbf{w})$ and $\boldsymbol{\epsilon}_\theta \approx \nabla_{\mathbf{u}, \mathbf{w}} \log p(\mathbf{u}, \mathbf{w})$ are essentially the same and the latter model is exactly DiffConPDE-lite, we will introduce DiffConPDE-lite first.

### B.4.1  DiffConPDE-lite

In general, DiffConPDE-lite follows the formulation of (Ho et al., 2020) which is also described in the main text. The data of $\mathbf{u}$ and $\mathbf{w}$ is fed in as images of size $(N_t, N_x)$ where $N_t$ is the number of time steps (11 and 10 respectively) and $N_x$ is the spatial grids (128). Since the two $N_t$s for $\mathbf{u}$ and $\mathbf{w}$ are inconsistent, we zero-pad them into the size of 16. Then, $\mathbf{u}$ and $\mathbf{w}$ are stacked as two channels and fed into the 2D DDPM model.

A 2D UNet $\boldsymbol{\epsilon}_\theta$ is used to learn to predict $\boldsymbol{\epsilon}$. It is structured into three main components: the downsampling encoder, the central module, and the upsampling decoder. The downsampling encoder is made up of four layers, each layer consisting of two ResNet blocks, one linear Attention block, and one downsampling convolution block. The central module also consists of two ResNet blocks and one linear Attention block. Each upsampling layer is the same as the downsampling layer except the downsampling block is replaced by the upsampling convolution block.

In our experiments, we found that the control result is best when learning the conditional probability distribution of $p(\mathbf{w}_{[0,T-1]}, \mathbf{u}_{[1,T-1]} \mid \mathbf{u}_0, \mathbf{u}_T)$ In summary, $\boldsymbol{\epsilon}_\theta$ takes in the current trajectory $\mathbf{u}$, control $\mathbf{w}$, step $k$, $\mathbf{u}_0$ and $\mathbf{u}_T$ as input, and predicts the noise of $\mathbf{u}$ and $\mathbf{w}$. Note that it is not trained to predict $\mathbf{u}_0$ and $\mathbf{u}_T$ which are used as a condition, but there are still model outputs at the corresponding locations for the data shape consistency across different design choices of DiffConPDE-lite.

### B.4.2 DIFFCONPDE

In terms of implementation, DiffConPDE is simply adding $\epsilon_\phi(\mathbf{w})$ to $\epsilon_\theta(\mathbf{u}, \mathbf{w})$ during inference as shown in Section 2.3, where $\epsilon_\theta$ is the output of the denoising network in DiffConPDE-lite while $\epsilon_\phi$ is a new denoising network that is trained to generate $\mathbf{w}$ following the dataset distribution. Therefore, we only describe the model of $\epsilon_\phi$ here.

$\epsilon_\phi$ takes input of $\mathbf{w}, k$ as in the standard DDPM and $\mathbf{u}_0, \mathbf{u}_T$ as guidance conditioning. The output of $\epsilon_\phi(\mathbf{w})$ is of the same shape as $\mathbf{w}$, so it can be treated as a network learning to sample from $p(\mathbf{w}) := \int p(\mathbf{u}, \mathbf{w}) \mathrm{d}\mathbf{u}$ The output of $\epsilon_\phi(\mathbf{w})$ at the locations of $\mathbf{u}$ is thus filled with zeros.

### B.5 TRAINING AND EVALUATION

**Training**  During training, the $\mathbf{u}_0$ and $\mathbf{u}_d$ without noise are fed into the model and the model outputs at the corresponding locations are excluded from the loss. In the partial observation settings, the unobserved data is invisible to the model during both training and testing as introduced in Appendix B.2. We simply pad zero in the corresponding locations of the model input and also exclude these locations in the training loss. Therefore, the model only learns the correlation between the observed states and control sequences. In the partial control setting, we train our DiffConPDE on the dataset with control being zero in $x \in [\frac{1}{4}, \frac{3}{4}]$. In this way, the model naturally learns to output zero at the "non-controllable" locations. We use the MSE loss to train the denoising UNets.

**Inference**  During inference, $\mathbf{u}_0$ and $\mathbf{u}_T$ are set to the target $\mathbf{u}_0$ and $\mathbf{u}_d$ so that the DDPM generates samples satisfying the PDE constraint that is also conditioned on the target ($\mathbf{u}_d$) or the constraint ($\mathbf{u}_0$). In the partial observation setting, the $\mathbf{u}_0$ and $\mathbf{u}_T$ drawn from the testing set are all filled zero at the unobserved locations $x \in [\frac{1}{4}, \frac{3}{4}]$, which is the same as the data used to train the UNets. In the partial control scenarios,

During inference, we replace the denoising network's output $\epsilon_\theta(\mathbf{u}, \mathbf{w})$ with $\epsilon_\theta(\mathbf{u}, \mathbf{w}) + (\gamma - 1)\epsilon_\phi(\mathbf{w})$. It is worth noting that $\epsilon_\theta(\mathbf{u}, \mathbf{w})$ denoises $u$ and $\mathbf{w}$ simultaneously while $\epsilon_\phi(\mathbf{w})$ only denoises $\mathbf{w}$. In our experiments, we found that adding a schedule to the output of the $\mathbf{w}$ network is beneficial. The results in Table 1 are generated with a reverse Sigmoid schedule following

$$\epsilon_k = \epsilon_\theta(\mathbf{u}_k, \mathbf{w}_k, k, \mathbf{u}_0, \mathbf{u}_T) + (\gamma - 1)\beta_{K-k}\epsilon_\phi(\mathbf{w}_k, k, \mathbf{u}_0, \mathbf{u}_T), \tag{17}$$

where $\beta$ is defined as the noise schedule in (Ho et al., 2020). The inference of DiffConPDE-lite is simply setting $\gamma = 1$, which neglects the effect of the model $\epsilon_\phi(\mathbf{w})$.

When trying to regulate the control energy, however, it is not as natural to learn a conditional model. Therefore, we use the external guidance $\mathcal{J} = \int \mathbf{w}(x, t) \mathrm{d}x \mathrm{d}t$ to produce cost-limited control sequences that are shown in Figure 2. The gradient of the external guidance is computed and added to $\epsilon_k$ in Eq. (17). Note that we use the predicted clean sample $\hat{\mathbf{w}}_k$ and $\hat{\mathbf{u}}_k$ at the $k$-th step to compute $\nabla \mathcal{J}$ since they suffer less from being noisy and leading to deviated guidance. $\mathbf{u}_k$ and $\mathbf{w}_k$ are computed following $\mathbf{x}_0 \approx \hat{\mathbf{x}}_k = \frac{1}{\sqrt{\bar{\alpha}_k}} \mathbf{x}_k - \frac{\sqrt{1-\bar{\alpha}_k}}{\sqrt{\bar{\alpha}_k}} \epsilon_k$ where $\mathbf{x}$ represents $\mathbf{u}$ or $\mathbf{w}$ as in Ho et al. (2020). In our experiments, we use a cosine scheduling of the external guidance, and thus the final predicted noise would be $\epsilon_k + \lambda \alpha_k \nabla \mathcal{J}$ where $\beta_k$ is the noise schedule in Ho et al. (2020) with the cosine schedule.

**Evaluation**  After generating the trajectory $\mathbf{u}$ and control sequence $\mathbf{w}$, we feed the control sequence $\mathbf{w}$ into the ground-truth solver and simulate the final state $\mathbf{u}$ given the generated $\mathbf{w}$ and the initial condition $\mathbf{u}_0$ directly drawn from the testing dataset. The solver is the same as the one used in data generation in Appendix B.1. Finally, we compute $J_{\text{actual}}$ following Eq. (12). In the partial observation setting, the MSE is computed only on the observed region, and in the control setting, the generated control will first be set to zero in the uncontrolled region before being fed into the solver.

## C  ADDITIONAL DETAILS FOR 2D JELLYFISH MOVEMENT CONTROL

### C.1  DATA GENERATION

We use the Lily-Pad simulator Weymouth (2015) to generate the training and testing dataset. The resolution of the 2D flow field is set to be $128 \times 128$. Actually, the flow field is assumed to be boundless in Lily-Pad. The head of the jellyfish is fixed at $(25.6, 64)$. Its two wings are represented by two identical ellipses, where the ratio between the shorter axis and the longer axis is 0.15. At each moment, the two wings are symmetric about the central horizontal line $y = 64$. For each wing, we sample $M = 20$ points along the wing to represent the boundary of the wing. The opening angle of the wings is defined as the angle between the longer axis of the upper wing and the horizontal line. It acts as the control sequence $\mathbf{w}$ in a 2D jellyfish control experiment.

Each trajectory starts from the largest opening angle and follows a cosine curve periodically with period $T' = 200$. Trajectories differ in initial angle, angle amplitude, and phase ratio $\tau$ (the ratio between the closing duration and a whole pitching duration). For each trajectory, the initial angle $\mathbf{w}_0$ is generated as follows: first, sample a random angle, called mean angle $\mathbf{w}^{(m)} \in [20°, 40°]$, then sample a random angle amplitude $\mathbf{w}^{(a)} \in [10°, \min(\mathbf{w}^{(m)}, 60° - \mathbf{w}^{(m)})]$. The initial $\mathbf{w}_0$ is set as $\mathbf{w}_0 = \mathbf{w}^{(m)} + \mathbf{w}^{(a)}$. The phase ratio $\tau$ is randomly sampled from $[0.2, 0.8]$. The opening angle $\mathbf{w}_t$ of step $t$ decreases from $\mathbf{w}^{(m)} + \mathbf{w}^{(a)}$ to $\mathbf{w}^{(m)} - \mathbf{w}^{(a)}$ as $t$ grows from 0 to $\tau T'$; then $\mathbf{w}_t$ increases from $\mathbf{w}^{(m)} - \mathbf{w}^{(a)}$ to $\mathbf{w}^{(m)} + \mathbf{w}^{(a)}$ as $t$ grows from $\tau T'$ to $T'$. Afterwards, $\mathbf{w}_t$ varies periodically for $t > T'$. The range of $\mathbf{w}_t$ is $[\mathbf{w}^{(m)} - \mathbf{w}^{(a)}, \mathbf{w}^{(m)} + \mathbf{w}^{(a)}] \subset [10°, 60°]$. This setting is similar to the study of the propulsive performance of jellyfish Kang et al. (2023). For each trajectory, we simulate for 600 simulation steps, i.e., 3 periods. To save space, we only save the piece of trajectory from $T' = 200$ to $3T' = 600$ steps with step size 10 because the simulation from $t = 0$ to $T' = 200$ is for initialization of the flow field. Then each trajectory is saved as a $\tilde{T} = (600 - 200)/10 = 40$ steps long sequence. An example of the simulated fluid field and the corresponding curve of opening angles are shown in Figure 6.

Besides the positions of the boundary points of wings and the opening angles $\mathbf{w}$, we also use another kind of image-like representation of the boundaries of wings as this representation contains spatial information that can be more effectively learned along with PDE states (fluid field) by convolution neural networks. For each trajectory, this image-like boundary representation is compatible with PDE states in shape. At each time step, boundaries of two wings are merged and then represented as a tensor of shape [3, 64, 64], where it has three features for each grid cell: a binary mask indicating whether the cell is inside a boundary (denoted by 1) or in the fluid (denoted by 0), and a relative position $(\Delta\mathbf{x}, \Delta y)$ between the cell center to the closest point on the boundary. For each trajectory, we save PDE states, opening angles, boundary points, boundary masks and offsets, and force data. They are specified as:

- PDE states $u$: shape $[\tilde{T}, 3, 64, 64]$. For each step, we save the states of the fluid field consisting of velocity in $\mathbf{x}$ and $y$ directions and pressure. To save space, we downsample the resolution from $128 \times 128$ to $64 \times 64$.
  - velocity: $[\tilde{T}, 2, 64, 64]$.
  - pressure: $[\tilde{T}, 1, 64, 64]$.
- opening angels $\mathbf{w}$: shape $[\tilde{T}]$. For each step, we save the opening angle in radians.
- boundary points: shape $[\tilde{T}, 2, M, 2]$. For each step, we save the boundary points on the upper and lower wings. Each wing consists of $M = 20$ points and each point consists of 2 coordinates. To make boundary points compatible with the downsampling of states, coordinates of x and $y$ directions are shrunk to half $(64/128)$ of the original values.
- boundary mask and offsets $b$: $[\tilde{T}, 3, 64, 64]$. For each step, we save the mask of merged wings with half coordinates of boundary points and offsets in both $\mathbf{x}$ and $y$ directions. The resolution is $64 \times 64$, compatible with that of the states.
  - mask: $[\tilde{T}, 1, 64, 64]$.
  - offsets: $[\tilde{T}, 2, 64, 64]$.

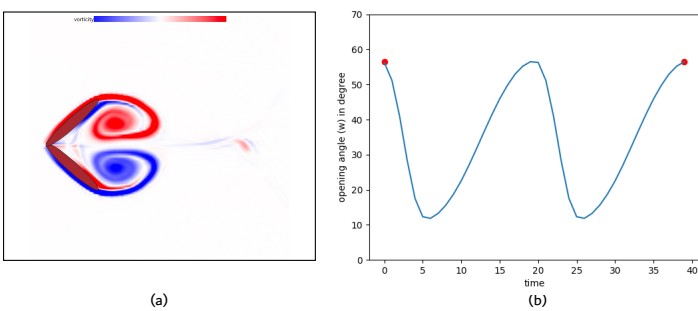

(a)            (b)

Figure 6: **Example of a flapping jellyfish in Lily-Pad simulator (a) and the corresponding curve of opening angle (b)**.

- force: shape $[\tilde{T}, 2]$. For each step, the simulator outputs the horizontal and vertical force from the fluid to the jellyfish. The horizontal force is regarded as a thrust to jellyfish if positive and a drag otherwise.

We generate $n = 30,000$ training trajectories and $n = 50$ testing trajectories. Trajectories differ in the above specified parameters $\mathbf{w}^{(a)}, \mathbf{w}^{(m)}$ and $\tau$. Each trajectory occupies about 2MB of storage and the total dataset costs about 100GB. To generate training samples, we use sliding time windows that contain $T = 20$ successive time steps of states and boundaries as a sample, which corresponds to $T^{'} = 200$ original simulation steps and constitutes exactly a period of wing movement. In this way, each trajectory can produce 20 samples. Therefore, we get 6 million training samples in total. In each training sample, the initial and the final time steps share the same opening angle due to periodicity, which serves as the conditions for control. For each test trajectory, we select the opening angle of the jellyfish in the initial time and the initial states as the control condition for both the initial and final time and state initial condition.

## C.2 EXPERIMENTAL SETTING

### C.2.1 FULL OBSERVATION

In this setting, we assume all the states of the fluid field are observable. That is, both the velocity of $x$ and $y$ directions and pressure are available in all the time steps of the training dataset and the initial time of the testing dataset.

### C.2.2 PARTIAL OBSERVATION

In this setting, we assume only partial states are observed. A typical scenario in fluid simulation and control is that we can only observe pressure data while the velocity data is not easy to access. That is, only pressure is available in all the time steps of the training samples and the initial time of the testing samples, hence the state tensor is of shape $[\tilde{T}, 1, 64, 64]$. Notice that even if only pressure is available, we can still compute the force of fluid on the jellyfish and consequently the control objective because force is fully determined by the shape of the jellyfish and pressure. The challenge of this partial observation setting is that the velocity variable $v$ is missing in Eq. (14), which makes the traditional numerical solver no longer applicable to solve this PDE control problem. However, this challenge could be well addressed by our method since it could learn the relationship between control and pressure despite missing of the velocity data, and use the accessible control objective as guidance for flapping control.

## C.3 MODEL

### C.3.1 ARCHITECTURE

We use a 3D U-Net as the backbone of our diffusion model. In this paper, the architecture of the 3D U-net we employed is inspired by Ho et al. (2022). To better capture temporal conditional dependencies, we modify the previous space-only 3D convolution into space-time 3D convolut

ion. Notably, we did not perform any scaling on the temporal dimension during downsampling or upsampling. Specifically, our U-net consists of three main modules: the downsampling encoder, the middle module, and the upsampling decoder. The downsampling encoder is composed of three layers, each incorporating two residual modules, one spatial attention module, one temporal attention module, and one downsampling module. The middle module consists of two residual modules, one spatial attention module, and one temporal attention module. Meanwhile, the upsampling decoder consists of four layers, each containing two residual modules, one spatial attention module, one temporal attention module, and one upsampling module. The input shape of our U-net is [batch size, frames, channels, height, width]. During convolution, the operation is performed on the [frames, height, width] dimensions. The output shape follows the same structure. Further details are provided in Table 3.

Table 3: **Hyperparameters of 3D-Unet architecture**.

| Hyperparameter name | Value |
| --- | --- |
| Kernel size of conv3d | (3, 3, 3) |
| Padding of conv3d | (1,1,1) |
| Stride of conv3d | (1,1,1) |
| Kernel size of downsampling | (1, 4, 4) |
| Padding of downsampling | (1, 2, 2) |
| Stride of downsampling | (0, 1, 1) |
| Kernel size of upsampling | (1, 4, 4) |
| Padding of upsampling | (1, 2, 2) |
| Stride of upsampling | (0, 1, 1) |
| attention heads | 4 |

### C.3.2 DIFFCONPDE-LITE

The DiffConPDE-lite method learns the denoising network of the joint distribution $p(\mathbf{u}, \mathbf{w}|\mathbf{c})$ where $\mathbf{u}$ is PDE states, $\mathbf{w}$ is the opening angle, and the conditions $\mathbf{c}$ consist of the initial angle $\mathbf{w}_0$, the initial state $\mathbf{u}_0$ and the final angle $\mathbf{w}_T = \mathbf{w}_0$. We adopt the 3D U-Net as the backbone. To make the opening angle (of shape $[T]$), align with PDE states (of shape $[T, 3, 64, 64]$ in full observation setting and $[T, 1, 64, 64]$ in partial observation setting) in shape, we expand the opening angle to shape $[T, 1, 64, 64]$ along spatial dimension by value copy. Besides, we also adopt the boundary mask and offsets representation, whose shape is $[T, 3, 64, 64]$, determined by the opening angles as an auxiliary model input because they contain explicit spatial features, which makes model learning more effective. Then states, boundary mask and offsets, and expanded opening angle are stacked along the channel dimension and we get a tensor of shape $[T, 7, 64, 64]$ in full observation setting or $[T, 5, 64, 64]$ in partial observation setting as the model input. The model output contains predicted noise of states and open angles. Thus its shape is $[T, 4, 64, 64]$ in the full observation setting or $[T, 2, 64, 64]$ in the partial observation setting, where the last channel corresponds to the predicted noise of opening angles and other channels correspond to predicted noise of states.

Table 4: **Hyperparameters of network architecture and training for the 2D experiment**.

| Hyperparameter name | full observation | partial observation |
| --- | --- | --- |
| Batch size | 16 | 16 |
| Optimizer | Adam | Adam |
| Learning rate | 0.0001 | 0.0001 |
| Loss function | MSE | MSE |

### C.3.3 DIFFCONPDE

DiffConPDE learns the denoising network of the joint distribution $p(\mathbf{u}, \mathbf{w}|\mathbf{c})$ and the marginal distribution $p(\mathbf{w}|\mathbf{c})$. The denoising network of $p(\mathbf{u}, \mathbf{w}|\mathbf{c})$ is exactly the same as the one introduced

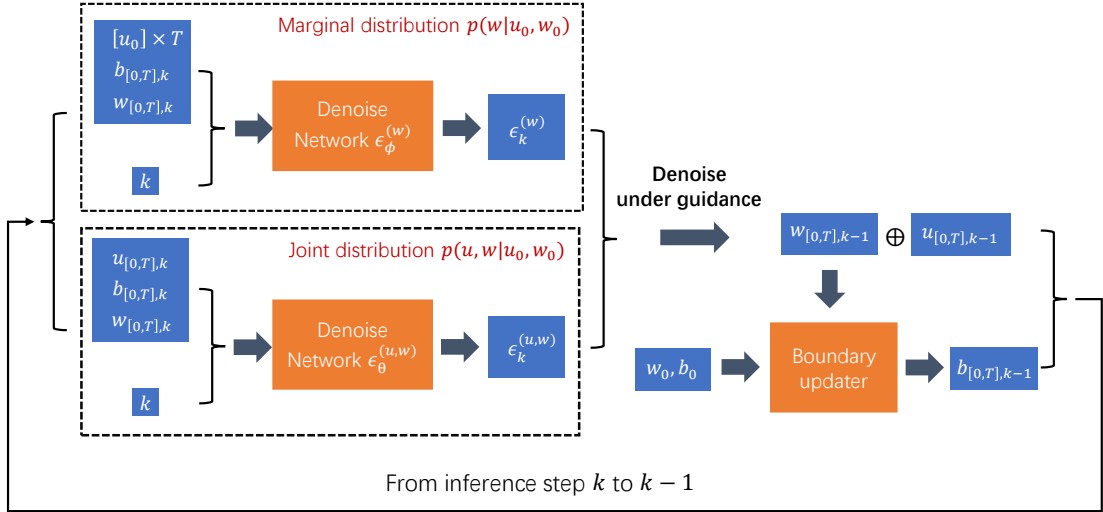

Figure 7: **Inference of our DiffConPDE in the 2D experiment.**

in the DiffConPDE-lite method. The denoising network of $p(\mathbf{w}|\mathbf{c})$ also adopts the 3D U-Net architecture. Its input size is the same as that of $p(\mathbf{u}, \mathbf{w}|\mathbf{c})$ in both full and partial observation settings. The difference is that the input states feature is replaced by the expansion of the initial state $\mathbf{u}_0$ along the time dimension by value copy. The output is the predicted noise of opening angles, whose shape is $[T, 1, 64, 64]$, no matter the full observation setting or partial observation setting.

### C.4 Training, Inference, and Evaluation

**Training.** We use the MSE (mean squared error) between model prediction and the Gaussian noise as the loss function. The batch size is chosen as 16 and the training involves 200,000 iterations. The learning rate starts from $1 \times 10^{-3}$ and multiplies a factor of at the 50000th and 150000th iterations. Training details are provided in Table 4.

**Inference.** The pipeline of inference is shown in Figure 7. Both diffused variables $\mathbf{u}_{[0,T]}$ and $\mathbf{w}_{[0,T]}$ are initialized from Gaussian prior and gradually denoised from denoising step $k = 1000$ to $k = 0$ based on denoising networks and guidance. Because we introduce the boundary mask and offsets as auxiliary inputs, the model input and output are not consistent in shape. Thus we introduce a surrogate model (shown as "Boundary updater" block in Figure 7) to update boundary mask and offsets $b_{[0,T],k}$ for each denoising $k$. Specifically, at each time step $t \in [0,T]$, $b_{t,k}$ is estimated by the initial boundary mask and offsets $b_0$, and the difference of opening angle $\mathbf{w}_{t,k} - \mathbf{w}_0$ from time step 0 to $t$, which is presented in the right part (after "Denoise under guidance") of Figure 7. Details about this surrogate model are presented in F.3. Notice that although this surrogate model is trained on noise-free data, we do not worry too much about its generalization to the noisy scalar $\mathbf{w}_{t,k}$ in inference because the estimated $\mathbf{w}_{t,k}$ does not deviate from the normalized range of noisy free $\mathbf{w}_t$ too much.

Our method introduces two kinds of inference: DiffConPDE-lite and DiffConPDE. In DiffConPDE-lite, we only use the denoising network of the joint distribution $p(\mathbf{u}, \mathbf{w}|\mathbf{u}_0, \mathbf{w}_0)$ for inference, while in DiffConPDE, we use the additional denoising network of the marginal distribution $p(\mathbf{w}|\mathbf{u}_0, \mathbf{w}_0)$ together with that of the joint distribution for inference. These two branches are plotted in the left part of Figure 7, where the notation $[\mathbf{u}_0] \times T$ means expand initial state $\mathbf{u}_0$ (of shape $[3, 64, 64]$) along time dimension by value copy to form a tensor of shape $[T, 3, 64, 64]$.

As for guidance, we use a surrogate force model to approximate the force of fluid on jellyfish. This model is detailed in Subsection F.2. In denoising step $k$, its input consists of two parts: the first one is the noise-free state $\hat{\mathbf{u}}_{[0,T],k}$ estimated from $\mathbf{u}_{[0,T],k}$; the second one is the noise-free boundary mask and offsets $\hat{b}_{[0,T],k}$ estimated from noise-free $\hat{\mathbf{w}}_{[0,T]}$ by the surrogate model to update boundaries, where $\hat{\mathbf{w}}_{[0,T]}$ is also estimated from $\mathbf{w}_{[0,T],k}$. The model output is force. Here we only use the

horizontal force. Notice that the force could be computed via the surrogate force model no matter whether states are fully or partial observation in that force is irrelevant to the velocity of the fluid. The control objective $\mathcal{J}$ in Eq. (15) is computed as a summation of force and $R(\hat{\mathbf{w}}_{[0,T]})$. We fix $\zeta = 1000$ as a default setting in Eq. (15) because this value can achieve a balance between scales of the average speed and the regularizer $R(\mathbf{w})$. We also study the Pareto performance of varying $\zeta$ in Table 2. Then the gradients of the objective $\mathcal{J}$ in terms of $\hat{\mathbf{u}}_{[0,T]}$ and $\hat{\mathbf{w}}_{[0,T]}$ are computed and used in guidance. For DiffConPDE-lite, these gradients are substracted from $[\mathbf{u}_{[0,T],k}, \mathbf{w}_{[0,T],k}]$ to generate $[\mathbf{u}_{[0,T],k-1}, \mathbf{w}_{[0,T],k-1}]$. For DiffConPDE, an additional term of noise $(\gamma - 1)\epsilon_\phi$ predicted from the denoising network of the marginal distribution $p(\mathbf{w}|\mathbf{u}_0, \mathbf{w}_0)$ should also be subtracted, as shown in the upper left part of Figure 7. The effect of this term is controlled by the scale of the hyperparameter $\gamma$.

**Evaluation.** The inference outputs opening angles $\mathbf{w}_{[0,T]}$ of $T = 20$ steps for 50 testing samples. In simulation, for each testing sample, the ground-truth first $T = 20$ steps of the opening angles (which corresponds to 200 simulation steps) are directly input to the Lily-Pad simulator for the reason of generating initial states $u_0$ of fluid, which is followed by the predicted control sequences of opening angles (interpolated to 200 steps of opening angles). The simulator outputs the horizontal force of fluid on the jellyfish for each simulation step. Finally, average speed $\bar{v}$, energy cost $R(\mathbf{w})$, and objective $\mathcal{J}$ are computed as metrics. The average speed $\bar{v} = \frac{1}{T} \int_0^T v_t \mathrm{d}t \approx v_0 + \frac{1}{T} \sum_{t=1}^{T-1} (T - t) F_t$ $\bar{v}$, where $v_0$ is the initial speed and $F_t$ is the horizontal thrust from the fluid. The mass of the jellyfish is assumed to be 1. The energy cost term $R(\mathbf{w}) = \sum_{t=1}^{T-1} (\mathbf{w}_{t+1} - \mathbf{w}_t)^2$, where the control sequence $\mathbf{w} = (\mathbf{w}_1, \cdots, \mathbf{w}_T)$ represents the predicted opening angles. The periodic term $d(\mathbf{w}_0, \mathbf{w}_T) = \max(|\mathbf{w}_T - \mathbf{w}_0| - \epsilon, 0)$ is the constraint of periodic opening angles with a small threshold $\epsilon = 0.01$.

# D    1D BASELINES

## D.1    PID

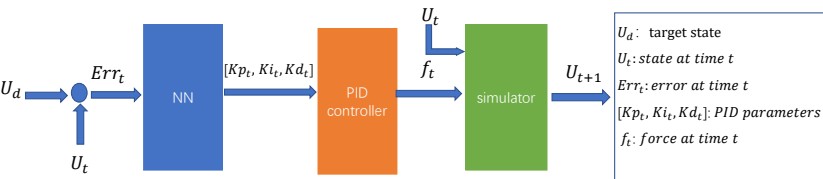

Figure 8: **The architecture of ANN PID.** To use MIMO PID controller to control $U_t$ to $U_d$, we train a neural-network-based PID parameter planner to output MIMO PID parameters based on $Err_t$, then use the PID controller to output the control sequence $f_t$.

Proportional Integral Derivative (PID) Li et al. (2006) control is a versatile and effective control method widely used in various real-world control scenarios. It operates by utilizing the difference (error) between the desired target and the current state of a system. PID control is often considered the go-to option for many control problems due to its simplicity and usefulness. However, despite its popularity, PID control does encounter certain challenges, such as parameter adaptation and limitations when applied to Single Input Single Output (SISO) systems. In our specific context, the 1D Burgers' Equation Control problem presents a Multiple Input Multiple Output (MIMO) control scenario, which makes it infeasible to directly employ PID control to regulate the Burgers' equation. Inspired by the early works Slama et al. (2019); Ding et al. (2022) using a neural network as a PID parameter adapter, we have integrated deep learning with PID control to tackle the MIMO control problem. As shown in Figure 8, ANN(artificial neural network) PID uses a neural network as a PID parameter adapter to output multiple sets of PID parameters and do multiple sets of SISO PID control.

The neural network to output PID parameters comprises two 1D convolutional layers, 2 fully connected layers, and 4 corresponding activation layers. We use the $L1$ loss of the current state and target state as training loss and the Adam optimizer Kingma & Ba (2014) to train the model. Detailed information can be found in Table 5.

Table 5: **Hyperparameters of network architecture and training for ANN PID**.

| Hyperparameter name | Full observation | Partial observation |
|---|---|---|
| Kernel size of conv1d | 3 | 3 |
| Padding of conv1d | 1 | 1 |
| Stride of conv1d | 1 | 1 |
| Activation function | Softsign | Softsign |
| Batch size | 16 | 16 |
| Optimizer | Adam | Adam |
| Learning rate | 0.0001 | 0.0001 |
| Loss function | MAE | MAE |

As PID itself is a SISO control method, ANN PID uses a neural network to get multiple sets of PID parameters to do multiple SISO PID controls for MIMO control in the Burgers' equation. But here, ANN PID requires the dimensions of inputs and outputs to be the same, so it can only cope with full observation, full control control problems, and partial observation, partial control control problems. Besides, the ANN PID controller has 2 training setups, including directly interacting with the solver and interacting with the 1D surrogate model in Appendix F.

## D.2    THE ADJOINT METHOD

The adjoint method Lions (1971); McNamara et al. (2004) is a mathematical technique used in the control problems of PDEs. It is widely utilized in many fields such as computational fluid dynamics, structural optimization, and machine learning.

The adjoint method first solves the forward problem, which is the original system of differential equations, to obtain the system's state as a function of the parameters. Then it constructs a Lagrangian function that includes the objective function, the forward problem's equations, and the associated boundary or initial conditions, all coupled together using the adjoint variables. We can derive adjoint equations by applying variational calculus, and solve the adjoint equations to get the adjoint variables. Finally, the gradients of the objective function with respect to the parameters, which are used to update the parameters, can be obtained by combining the solutions of the forward and adjoint problems using the adjoint variables.

Our implementation follows the previous work Hwang et al. (2022). While the use of the adjoint method requires the form of the equations, it can be only applied to the 1D Burgers' equation control. Also, it cannot handle cases where states of systems are partial observation. For the partial control setting, we set the output control sequences of the adjoint method to zero where the control is not allowed to be applied. We choose 10 and 100 numerical time steps for the adjoint method. Accordingly, the discrete computation of energy takes $dt = 0.1$ and $dt = 0.01$.

## D.3    SAC

The Soft Actor-Critic (SAC) algorithm Haarnoja et al. (2018) is a cutting-edge reinforcement learning method. Conceptualized as an improvement over traditional Actor-Critic methods, SAC distinguishes itself by introducing an entropy regularization term into the loss function, which encourages the policy to explore more efficiently by maximizing both the expected cumulative reward and the entropy of the policy itself.

Compared with Deep Deterministic Policy Gradient (DDPG) algorithm Lillicrap et al. (2015); Pan et al. (2018), SAC's entropy regularization encourages more effective exploration and prevents early convergence to suboptimal policies, a limitation often seen with DDPG's deterministic approach. Additionally, SAC's twin Q-networks mitigate the overestimation bias that can affect DDPG's value updates, leading to more stable learning. The automatic tuning of the temperature parameter in SAC further simplifies the delicate balance between exploration and exploitation, reducing the need for meticulous hyperparameter adjustments. Consequently, these features render SAC generally more sample-efficient and robust, particularly in complex and continuous action spaces.

Table 6: **Hyperparameters of 1D SAC.** The full observation partial control, partial observation full control, and partial observation partial control settings share the same hyperparameters.

| Hyperparameter name | Value |
|---|---|
| Hyperparameters for 1D Burgers' equation control: | |
| Discount factor for reward | 0.5 |
| Target smoothing coefficient | 0.05 |
| Learning rate of critic loss | 0.0003 |
| Learning rate of entropy loss | 0.003 |
| Learning rate of policy loss | 0.003 |
| Training batch size | 8192 |
| Number of episodes | 1500 |
| Number of model updates per simulator step | 50 |
| Value target updates per step | 15 |
| Size of replay buffer | 1000000 |
| Number of trajectories interacted with the environment per step | 1 |
| Number of layers of critic networks | 3 |
| Number of hidden dimensions of critic networks | 4096 |
| Number of layers of the policy network | 5 |
| Number of hidden dimensions of the policy network | 4096 |
| Activation function | ReLU |
| Clipping's range of policy network's standard deviation output | $\left[e^{-20}, e^2\right]$ |

During training, experience for training is stored in a replay buffer and sampled randomly to update the networks. All data in the training set are in the replay buffer at the beginning. For offline SAC, the replay buffer is unchanged, while online SAC alternates between collecting experience by interacting with the environment and updating the networks with the replay buffer. And offline SAC only uses the surrogate model trained with the training set instead of the real environment to collect experience. The policy network is updated to maximize the expected return, considering both the Q-value and the entropy term. The critic networks are updated to minimize the distance between their Q-value predictions and the target Q-values. SAC also employs a target critic network for the critic networks, which are slowly updated with the weights of the main critic network to stabilize training. For the inference, SAC uses the policy network to determine the action by selecting the action with the highest probability.

In practice, to help the system approximate the target state accurately and quickly, we need to include the distance between the states of every time step and the target state in the reward. So the reward function of time step $t$, state $u_t$, target state $u_T$ and action $\mathbf{w}_t$ here is defined as

$$r(t, \mathbf{u}_t, y_T, \mathbf{w}_t) = -\int_{\Omega} |\mathbf{u}_t - \mathbf{u}_d|^2 \mathrm{d}\mathbf{x} - \alpha \int_{\Omega} |\mathbf{w}_t|^2 \mathrm{d}\mathbf{x},$$

where $\Omega$ is the space domain and $\alpha$ is the weight of energy. We take the Adam optimizer Kingma & Ba (2014) to train the networks and update the temperature parameter. The detailed values of hyperparameters are provided in Table 6.

### D.4 SUPERVISED LEARNING

The paper Hwang et al. (2022) proposes a supervised-learning-based control algorithm that takes a neural operator as a surrogate model to solve control problems. It contains two stages. In the first stage, we take a neural operator to learn the PDE constraint as Appendix F. The three CNNs respectively reconstruct $u$, reconstruct $\mathbf{w}$ and learn the transition from $u_t$ to $u_{t+1}$. More details are in Appendix F. In the second stage, these three neural networks are used as surrogate models to calculate the gradient of the objective function with respect to the control input. We consider the control $\mathbf{w}$ as a learnable parameter and update it with the gradient.

To enhance the accuracy, we adopt the LBFGS optimizer Liu & Nocedal (1989), which is more accurate while slower than the Adam optimizer. We record the hyperparameters of the second stage in Table 7.

Table 7: **Hyperparameters of the second stage of the 1D supervised learning method**.

| Hyperparameter name | Value |
|---|---|
| Hyperparameters for 1D Burgers' equation control: (full observation partial control) | |
| Learning rate of $\mathbf{w}$ updating | 0.1 |
| Number of epochs | 300 |
| Weight of objective function loss | 500 |
| Weight of reconstruction loss | 0.03 |
| Termination tolerance on first order optimality of LBFGS optimizer | $4 \times 10^{-7}$ |
| Termination tolerance on parameter changes LBFGS optimizer | $4 \times 10^{-7}$ |
| Hyperparameter name | Value |
| Hyperparameters for 1D Burgers' equation control: (partial observation partial/full control) | |
| Learning rate of $\mathbf{w}$ updating | 0.1 |
| Number of epochs | 300 |
| Weight of objective function loss | 50000 |
| Weight of reconstruction loss | 3 |
| Termination tolerance on first order optimality of LBFGS optimizer | $4 \times 10^{-7}$ |
| Termination tolerance on parameter changes LBFGS optimizer | $4 \times 10^{-7}$ |

## E    2D BASELINES

### E.1    MPC AND SL

Model Predictive Control (MPC) Schwenzer et al. (2021) is a control strategy that solves an optimization problem repeatedly to determine the optimal control inputs for a dynamic system. It operates over a finite prediction horizon, optimizing a cost function and applying only the first control action. In the 2D jellyfish movement control problem, MPC uses the control sequences $\mathbf{w}$ and the fluid states as internal state variables. Without the need to train a control agent model, MPC relies on 2D surrogate models mentioned in Appendix F to estimate future states based on the current state and control. We use backpropagation to compute the gradient, update the control action sequences, and optimize the control objective $\mathcal{J}$ in Eq. (15). For every time step, we get the optimized sequences from this time step forward in this way, and only the first control sequence of the optimized control sequences is applied. Compared with MPC, the Supervised learning (SL) method Hwang et al. (2022) only optimizes the entire control sequences and employs the entire sequences.

MPC is an optimization technique that aims to optimize the control of complex dynamic systems by considering future predictions. This approach can optimize performance measures over a future time horizon, handle systems with multiple variables and constraints, adapt to changes in the system behavior, and offer good performance even in the presence of nonlinearity. Nevertheless, using MPC comes with a high cost, both in terms of computational resources and time. Additionally, adapting the optimization hyperparameters for MPC can be a challenging task. In our experiment, both MPC and SL face difficulties when trying to generate smooth opening angle control curves, even when constraints $R(\hat{\mathbf{w}})$ are included in the optimization objective $\mathcal{J}$. In the case of multi-objective optimization problems, it becomes even more challenging for them to simultaneously achieve both the higher speed (bigger $\bar{v}$) and the control curves smoothness (smaller $R(\hat{\mathbf{w}})$).

### E.2    SAC

For the 2D case, the algorithm and basic architecture of SAC are the same as the 1D case in Appendix D.3. When designing the reward function for the 2D jellyfish movement control, we find the periodic condition of the opening angle curves is hard to constrain. So to satisfy the periodic condition better, we include the distance between $\mathbf{w}_t$ of every time step $t$ and $\mathbf{w}_0$. Also, we both consider the squared and absolute error of $(\mathbf{w}_t - \mathbf{w}_0)$ since they respectively constrain the periodic condition when $(\mathbf{w}_t - \mathbf{w}_0)$ is large and small. As a result, the reward function of time step $t$, force $F_t$, opening angle $(\mathbf{w}_{t-1}, \mathbf{w}_t)$ and condition angle $\mathbf{w}_0$ is defined as

$$r(t, \mathbf{w}_{t-1}, \mathbf{w}_t, \mathbf{w}_0) = (T - t) * F_t - \lambda_1 (\mathbf{w}_t - \mathbf{w}_{t-1})^2 - \lambda_2 ((\mathbf{w}_t - \mathbf{w}_0)^2 + |\mathbf{w}_t - \mathbf{w}_0|),$$

where $\lambda_1$, $\lambda_2$ are weights of different terms. We take the Adam optimizer Kingma & Ba (2014) to update the weights of networks and the temperature parameter as in the 1D experiment. The hyperparameters are reported in Table 8. In particular, we take the best checkpoint to evaluate the final performance, thus the actual number of training episodes for each setting ranges from about 100 to about 300.

Table 8: **Hyperparameters of 2D SAC.** The full observation and partial observation settings share the same hyperparameters.

| Hyperparameter name | Value |
|---|---|
| Hyperparameters for 2D Jellyfish movement control: | |
| Weight of the constraint of periodic condition $\beta$ | 0.001 |
| Discount factor for reward | 0.5 |
| Target smoothing coefficient | 0.05 |
| Learning rate of critic loss | 0.0003 |
| Learning rate of entropy loss | 0.0003 |
| Learning rate of policy loss | 0.0003 |
| Training batch size | 2048 |
| Number of episodes | 350 |
| Number of model updates per simulator step | 20 |
| Value target updates per step | 15 |
| Size of replay buffer | 11400001 |
| Number of trajectories interacted with the environment per step | 5 |
| Activation function | ELU |
| Clipping's range of policy network's standard deviation output | $\left[e^{-5}, e^{-2}\right]$ |

## F  SURROGATE MODELS

### F.1  1D SURROGATE MODEL

For the control problem of 1D Burgers' equation, our 1D surrogate model is based on the previous paper Hwang et al. (2022), which uses 2 autoencoders to model dynamics in the latent space. The neural simulator architecture and training details are shown in Table 9.

Table 9: **Hyperparameters of 1D surrogate model**.

| Hyperparameter name | Full observation, partial control | Partial observation, full control | Partial observation, partial control |
|---|---|---|---|
| Autoencoder of state | | | |
| Convolution kernel size | 5 | 5 | 5 |
| Convolution padding | 2 | 2 | 2 |
| Activation function | ELU | ELU | ELU |
| Latent vector size | 256 | 128 | 128 |
| Autoencoder of force | | | |
| Convolution kernel size | 5 | 5 | 5 |
| Convolution padding | 2 | 2 | 2 |
| Activation function | ELU | ELU | ELU |
| Latent vector size | 256 | 256 | 256 |
| Training | | | |
| Training batch size | 5100 | 5100 | 5100 |
| Optimizer | Adam | Adam | Adam |
| Learning rate | 1e-3 | 1e-3 | 1e-3 |
| Training epochs | 500 | 500 | 500 |
| Learning rate scheduler | cosine annealing | cosine annealing | cosine annealing |

### F.2  2D FORCE MODELS

**Dataset.** In 2D jellyfish movement control experiments, we train a force surrogate to approximate the computation of the average speed of the jellyfish for the guidance of inference, which is implemented by a neural network and is thus differentiable. The training data consists of pressure, boundary, and force data in the training trajectories. Each training trajectory amounts to $\tilde{T} = 40$ training samples. Therefore, we have 1.2 million training samples and 4 thousand testing samples in total.

**Model.** The model's input contains pressure, boundary mask, and offsets with shape $4 \times 64 \times 64$ at a certain time step, and the output is the corresponding forces of $x$ and $y$ directions. The model architecture is the down-sampling part of a U-Net Ronneberger et al. (2015) that embeds the input features into a 512-dimensional hidden representation; then we use a linear function with output dimension two to output forces.

**Training.** We use MSE (mean squared error) loss between the ground truth and predicted forces to train the force surrogate model. The optimizer is Adam (Kingma & Ba, 2014). The batch size is 64. The model is trained for 10 epochs. The learning rate starts from $1 \times 10^{-4}$ and multiplies a factor of 0.1 every three epochs. After training, the relative $l_2$ test error is 0.4%.

### F.3  2D BOUNDARY MASK AND OFFSETS UPDATER

**Dataset.** In 2D jellyfish movement control experiments, we train a boundary mask and offsets updater surrogate to approximate the transition of boundary mask and offsets from time step $0$ to $t$. Thus each training trajectory amounts to $\tilde{T} - 1 = 39$ training samples.

**Model.** The input is the boundary mask and offsets at time step $0$ with shape $3 \times 64 \times 64$, and the difference of the opening angle from $0$ to $t$. The output is the boundary mask and offsets at time step $t$ with shape $3 \times 64 \times 64$. The model architecture is the U-Net Ronneberger et al. (2015) with additional scalar input, similar to the denoising network in DDPM Ho et al. (2020), where the input scalar diffusion step is replaced by the angle difference in our model.

**Training.** We use MSE (mean squared error) loss between the ground truth and predicted boundary mask and offsets to train this surrogate model. Hyperparameters of training are the same as those of the force surrogate model.

### F.4  2D SIMULATOR

**Dataset.** In 2D jellyfish movement control experiments, we need to train a surrogate model as a solver of the PDE for the baseline methods like SAC (online) and MPC, because of their iterative nature. Conversely, our diffusion method *does not* need this surrogate model. This model approximates the transition of states under the boundary condition from time step $t$ to $t + 1$. Thus each training trajectory amounts to $\tilde{T} - 1 = 39$ training samples. We train two versions of this model for full/partial observation settings.

**Model.** This surrogate model is also implemented by the U-Net Ronneberger et al. (2015). The model input is the states, boundary mask, and offsets at time $t$ with shape $6 \times 64 \times 64$ for the full observation setting and $4 \times 64 \times 64$ for the partial observation setting. The output is the predicted states at time step $t + 1$, with shape $3 \times 64 \times 64$ for the full observation setting and $1 \times 64 \times 64$ for the partial observation setting.

**Training.** We use MSE loss between the ground truth and predicted states to train this surrogate model. Hyperparameters of training are the same as those of the force surrogate model.

## G  EFFECT OF HYPERPARAMETER

Performance of DiffConPDE is determined by the hyperparameter $\gamma$. Since diffusion models denoise gradually, we use a varying sequence of $\gamma = \{\gamma_k\}_{k=1}^{K}$ to substract prior in DiffConPDE. Specifically, the schedule of $\gamma$ is set as $\gamma_k = 1 - \xi \cdot \beta_{K-k}, k = 1, \cdots, K$, where $\xi$ is a fixed coefficient to control the scale of $\gamma$ and $\beta = \{\beta_k\}_{k=0}^{K}$ is the schedule of variances of noise in DDPM Ho et al. (2020). In our implementation, we use the sigmoid schedule of $\beta$ Jabri et al. (2022). The total number of

Table 10: **Results of different $\gamma$ on 2D jellyfish movement control**.

| $\gamma_1$ | $\xi$ | Average speed ($\bar{v}$) | $R(\mathbf{w})$ | objective $\mathcal{J}$ |
|---|---|---|---|---|
| 0.6 | 0.4 | 410.6 | 0.2581 | -152.51 |
| 0.7 | 0.3 | 279.87 | 0.2058 | -74.11 |
| 0.8 | 0.2 | 197.18 | 0.1312 | -65.99 |
| 0.9 | 0.1 | 76.97 | 0.0741 | -2.84 |
| 1.0 | 0 | 95.04 | 0.0746 | -20.47 |
| 1.1 | -0.1 | 81.41 | 0.0742 | -7.21 |
| 1.2 | -0.2 | 84.56 | 0.0736 | -10.93 |
| 1.3 | -0.3 | 65.12 | 0.0725 | 7.38 |
| 1.4 | -0.4 | 65.02 | 0.0734 | 8.43 |
| 1.5 | -0.5 | 64.07 | 0.0752 | 11.1 |

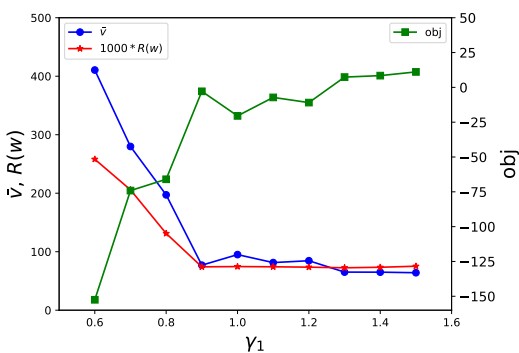

Figure 9: **Results of different $\gamma$ on 2D jellyfish movement control**.

inference steps is $K = 1000$. Thus we only need to tune $\xi$ to examine the effect of $\gamma$. When $\xi < 0$, DiffConPDE is prone to restrict $\mathbf{w}$ within its prior distribution of training dataset in inference. When $\xi > 0$, DiffConPDE is more likely to generate new kinds of $\mathbf{w}$ beyond training ones. When $\xi = 0$, DiffConPDE degenerates to DiffConPDE-lite. In 2D experiments, We set default $\xi = 0.3$ and the corresponding $\gamma_1 = 0.7$ as we empirically find this value performs well and steadily. We present the performance of DiffConPDE on the 2D jellyfish movement control task under different $\gamma$ in Figure 9 and Table 10. We can observe that the performance increases along with decreasing of $\gamma_1$. When $\gamma_1 < 0.6$, invalid generated control sequences emerge because the prior is largely overlooked. Thus the valid interval for $\gamma$ of prior reweighting on this task is $[0.6, 1.0]$. It is interesting to find that when $\gamma_1 > 1$, the performance decreases. This may be caused by the strict constraint of the prior distribution of $p(\mathbf{w}, \mathbf{c})$, which results in generating control sequences similar to those from training datasets and thus not good.

## H  LIMITATION AND FUTURE WORK

Although DiffConPDE solves the PDE control problems with outstanding performance, there are still several limitations that provide exciting opportunity for future works. Firstly, the inference process of the diffusion model, the foundational component of DiffConPDE, may benefit from enhanced efficiency by using e.g. distillation Salimans & Ho (2022); Li et al. (2023). Secondly, the training of DiffConPDE is currently conducted in an offline fashion, lacking interaction with a ground-truth solver. Incorporating solvers into the training framework could facilitate real-time feedback, enabling the model to adapt dynamically to the environment and discover novel strategies and solutions. Furthermore, our proposed DiffConPDE presently operates in an open-loop manner, as it does not consider real-time feedback from solvers. Integrating such feedback would empower the algorithm to adjust its control decisions for the subsequent steps based on the evolving state of the environment.

Moving forward, we plan to expedite the inference process by drawing insights from relevant prior works, such as Salimans & Ho (2021); Meng et al. (2023); Li et al. (2023). Additionally, we aim to

build upon our existing trajectory knowledge to generalize future trajectories. Furthermore, we intend to incorporate solver interactions into our training methodology, enhancing the model's adaptability and effectiveness.

