# OpenReview forum: "Generative PDE Control"
_ICLR.cc/2024/Workshop/AI4DiffEqtnsInSci — AI4DiffEqtnsInSci @ ICLR 2024 Oral_

### Official Review · Reviewer_sEM8 · 2024-02-21
**Strong paper with strong empirical results, possible spotlight**

**Rating:** 9
**Confidence:** 4

**Review:**

# Summary
This work explores the use of diffusion models in the setting of controlled partial differential equation (PDE). In this setting, the goal is to minimize a control objective $\mathcal J (\mathbf u, \mathbf w)$ that is a function of both the trajectory of the system $\mathbf u$ and the external control signal $\mathbf w$. The dynamics of the trajectory $\mathbf u$ are governed by a PDE system that depends on the control signal $\mathbf w$ and initial and boundary conditions $\mathcal c$.

The authors propose that there is a generative model of $p(u, w \vert c)$. The negative log-density of this distribution is known as a parameterized energy-based model. The controlled PDE optimization problem can be recast as minimizing the negative log-density of $p(u, w \vert c)$ with $\mathcal J(u, w)$ as an added regularization term (Equation 5).

To solve the new optimization problem, the authors first train a diffusion model to learn $p(u, w \vert c)$. The key observation is that the original diffusion model presented in [1] yields an estimate of the gradient of $E_{\theta}$ at each step (resembling Langevin dynamics). With this observation, the authors modify the standard sampling step of a diffusion model to incorporate the gradient of the control objective $\mathcal J$. This leads to an optimization routine that targets the desired objective.

The authors also propose a scheme for minimizing the impact of choice of prior on the control sequences $p(w \vert c)$. This works by tempering the density of $p(w \vert c)^{\gamma}$ (I believe there was a typo in the text that I have corrected here). Similar to the $\beta$ parameter in the original diffusion paper, $\gamma$ also changes with each step of the diffusion. Specifically, $\gamma_{1} = 0.7$ and $\gamma_{K} = 1$, so the "prior component" of the target distribution becomes less diffuse in later steps of the diffusion.

The authors present results on a 2-d jellyfish control task in the main body. Their proposed method outperforms all other competing methods. The optimal strategy their method found was corroborated in other works as being an effective strategy for jellyfish movement.

# Review

Overall, with an interesting method and strong empirical results, this is a nice paper that should be a strong acceptance to the workshop. In fact, I believe this would be a good candidate for a spotlight paper, since it demonstrates a nice, effective use of deep learning to solve real challenges in numerical methods.

## Originality
This paper builds upon well-established diffusion models to facilitate controlled PDE optimization. [1] commented that the diffusion sampling process resembles Langevin dynamics and follows the gradient of the log probability density. This work exploits that fact by  modifying the diffusion sampling algorithm to incorporate the gradient of the control objective $\mathcal J$. To my knowledge, this is a new idea, though I am not deeply familiar with the diffusion model literature.

## Quality
The overall quality of the work is high for a workshop paper. The method is well-explained, and the empirical evaluation is comprehensive for a workshop paper. The empirical evaluation clearly demonstrates the effectiveness of the proposed method at an assigned task.

While the quality is high for a workshop paper, I have some remarks below that should be answered in a full paper.

1. The probability distribution of $u, w$ needs to be more carefully defined and characterized. The set $\{u: \mathcal C(u, w) = 0 \}$ will have Lebesgue measure 0 on the space where $u$ is defined ($[0, \mathcal{T}] \times \Omega$). As a result, the density of $p(u \vert w, c)$ is a point mass on this space. It does not seem that this has caused any practical issues in the current empirical setting. However, samples from the trained diffusion model almost certainly do not satisfy $\mathcal C(u, w)$, and this could pose problems in situations where the ground truth simulator is not available.

2. I would like to see more theoretical results about the convergence of the control optimization to a valid solution. From my comment above (1), the solution yielded by algorithm (1) will almost certainly not satisfy $C(u, w) = 0$. It would be beneficial to know the conditions under which the solution is arbitrarily close to satisfying $C(u, w) = 0$, even if these conditions are not met in practice. Knowing the failure modes of this methodology is important for knowing the limitations and addressing them in future work.

## Clarity
The writing is clear throughout. The visuals of the experiment are quite nice. A few concepts, such as the prior re-weighting scheme, required me to reference the appendix to get a full understanding. Specifically, in the main text, it is not made clear that $\gamma$ depends on the index $k$, similar to the parameter $\beta_{k}$. However, this is likely a consequence of the 4-page limit.

## Significance
This work provides a blueprint on how deep learning can solve real problems better than classical methods, which embodies the spirit of this workshop. I also believe that this approach could be useful in a broad range of applications, even beyond controlled PDES.

## General Feedback
Currently, you train the diffusion model to estimate $E_{\theta}$ by having it target the distribution with density proportional to $\exp(-E_{\theta}(u, w, c))$. What if, instead, you trained the diffusion model to target the distribution with density proportional to $\exp(-E_{\theta}(u, w, c) - \lambda \mathcal J(u, w))$? This will lead to more weight on samples with a lower $\mathcal J$.

---

### Official Review · Reviewer_BXuX · 2024-02-26
**Good paper, clear accept, low confidence**

**Rating:** 8
**Confidence:** 2

**Review:**

Summary: This paper introduces an approach to PDE control problems; the goal is to learn to control the PDE behavior. The DiffConPDE model learns under the condition that the explicit form of the PDE may not be known, rather what is known is observed control sequences and trajectory data. A generative denoising diffusion model is used to generate the trajectory, and an inference algorithm estimates the optimal control signal.

My research area is in solving PDEs with ML, not PDE control. I also am not familiar with the mathematics of generative diffusion models (sorry!), so I do not understand the meaning of equations (6) (9) and (10). For those reasons, my review comes with relatively low confidence.

This paper is well-written, well-motivated, clearly of interest to the workshop, and seems to achieve good results (both in the main text and the appendix). I am quite happy with this paper.

It is possible that there are flaws with this paper that I am not aware of (better baselines, prior work with similar ideas, etc). However, I don't have any reason to suspect these flaws exist.

I believe this paper is clearly in the top 50% of accepted papers. I am not familiar enough with this research area to decide that this paper falls into the top 15% of accepted papers, but I would not disagree if someone more familiar with the research area felt that it deserved such a rating.

---

### Meta-Review · Area_Chair_qQSn · 2024-02-26

**Recommendation:** Accept (Oral)

**Metareview:**

The reviewers both clearly indicate that this paper is a clear accept. It proposes to use diffusion models for PDE Control and tests both 1D and 2D examples. The topic is very relevant to the workshop and I vote for acceptnace.

---

### Decision · Program_Chairs · 2024-02-29

Accept (Oral)